# Changes in Key Biomechanical Parameters According to the Expertise Level in Runners at Different Running Speeds

**DOI:** 10.3390/bioengineering9110616

**Published:** 2022-10-26

**Authors:** Cagla Fadillioglu, Felix Möhler, Marcel Reuter, Thorsten Stein

**Affiliations:** 1BioMotion Center, Institute of Sports and Sports Science (IfSS), Karlsruhe Institute of Technology, 76131 Karlsruhe, Germany; 2Department of Applied Training Science, German University of Applied Sciences for Prevention and Health Management (DHfPG), 66123 Saarbrücken, Germany

**Keywords:** running economy, running style, duty factor, vertical oscillation, stride frequency

## Abstract

Running has become increasingly popular worldwide. Among runners, there exists a wide range of expertise levels. Investigating the differences between runners at two extreme levels, that is novices and experts, is crucial to understand the changes that occur as a result of multiple years of training. Vertical oscillation of center of mass (CoM), stride frequency normalized to the leg length, and duty factor, which describes the step time relative to the flight time, are key biomechanical parameters that have been shown to be closely related to the running economy and are used to characterize the running style. The variability characteristics of these parameters may reveal valuable information concerning the control of human locomotion. However, how the expertise level and running speed affect the variability of these key biomechanical parameters has not yet been investigated. The aim of this study was to analyze the effects of expertise level (novice vs. expert) and running speed (10 km/h vs. 15 km/h) on these parameters and their variability. It was hypothesized that expert runners would have lower vertical oscillation of CoM, normalized stride frequency, and duty factor and show less variability in these parameters. The parameters’ variability was operationalized by the coefficient of variation. The mean values and variability of these key biomechanical parameters according to expertise level and running speed were compared with rmANOVAs. The results showed that the experts had a lower duty factor and less variable vertical oscillation of CoM and normalized stride frequency, independently of the running speed. At a higher running speed, the variability of vertical oscillation of CoM was higher, whereas that of normalized stride frequency and duty factor did not change significantly. To the best of our knowledge, this is the first study analyzing the effects of expertise level and running speed on the variability of key biomechanical parameters.

## 1. Introduction

Running is a sport that has been growing in popularity over the years [1]. Through multiple years of training, runners accomplish a variety of changes at different levels that include metabolic, neuromuscular, and biomechanical efficiency and ultimately result in improved running economy [2]. The changes in running economy play an important role in running performance, since a smaller energy expenditure at a given speed is beneficial, especially in disciplines that require running with submaximal speed for long distances. Therefore, not only cardiovascular and metabolic fitness (e.g., VO_2_max) but also biomechanical and neuromuscular efficiency are crucial for individual running economy [2]. Even though VO_2_max is typically used as a measure of running economy [3,4], studies that track the performance development of elite athletes for several years have reported improvements in performance without significant changes in VO_2_max [5,6]. This emphasizes the importance of considering the whole scope of variables that are associated with running economy. Apart from factors such as age, gender, body temperature, and muscle fiber distribution, several biomechanical parameters as well as the running style have been shown to influence the running economy [7,8].

Positive correlations have been observed between isolated parameters and running economy. In particular, the vertical oscillation of the center of mass (osc_CoM) [2,8,9,10], step length (SL) [2,10], and step frequency (SF) [3,10] were identified as factors that influence the running economy in various running-related studies. The findings of these studies have indicated that an increase in running economy is associated with a lower osc_CoM [2,8,9,11] and a lower SF [3,10], whereas the results for SL have not been consistent across studies [10,12]. These varying results can be explained by the self-optimized SL/SF ratio [13,14]. On this basis, analyzing different running styles may be preferable to a comparison of isolated parameters when trying to understand the kinematic adaptations that occur as a result of training. However, the parameters used for the operationalization of the running style vary across the literature. Recently, in a study by van Overen et al. [13], it was suggested that the duty factor (DF), which is the ratio of stance to stride time, together with the SF normalized to the leg length (SF_norm) would be suitable to describe the running style and can be used for comparisons between individuals.

The expertise level is a major factor that influences various biomechanical parameters, including spatio-temporal parameters. Therefore, in studies that attempt to distinguish between groups of runners in terms of their expertise level and define them, several terms are used, including “expert”, “novice”, “elite”, “amateur”, and “good”. In this study, the terms expert (to refer to elite runners), good, and novice (to refer to amateur runners) will be used to avoid confusion between terminologies. Such studies have reported that SF was found to increase with increase in experience, stride time was found to be unaffected [15,16,17,18], and DF was found to decrease [19]. In addition, the effects of running experience on running economy have been reported in various studies. Importantly, experts were shown to have a more efficient running style than novices [9,20]. In studies that compared the performance of expert and good distance runners, it was found that expert runners have a slightly lower osc_CoM and a better running economy than good runners [2]. 

Apart from the running experience, the running speed was also shown to influence the kinematic variables. Padulo et al. [21] showed that both for expert and for novice runners, SL and SF increased, while stance time decreased with an increase in running speed. A study by García-Pinillos et al. [22] investigated the alterations in spatiotemporal parameters in novice endurance runners and showed that stance time becomes shorter, while SL and flight time become longer, as speed increases. Furthermore, not only the mean values but also the variability in spatiotemporal parameters was found to change with an increasing speed. Jordan et al. [23] suggested that the variability in running gait is not random but manifests self-similarity depending on the gait speed and therefore, it may help to understand the control of human locomotion.

The term “movement variability” can refer to different aspects according to the context [24]. In sports, the main way to achieve the desired goal (e.g., hitting the basketball hoop) is to achieve consistency over multiple repetitions. On the other hand, a certain level of variability may increase the flexibility of the whole biological system and ultimately help to adapt to different conditions, such as fatigue and environmental changes, and may also help to reduce injuries [24]. It was suggested that the variability of a parameter, which is important as a movement goal, can be minimized over several movement repetitions, and thereby, variations in less important parameters may be allowed to a certain level so that their co-variation establishes a flexible but stable system [25,26,27]. However, in the case of sports such as running, it is difficult to definitively determine which parameters are prioritized by the central nervous system (CNS) in terms of control. Typically, runners aim to run as fast as possible for a given distance and, thereby, try to improve their running style to make it more energy-efficient [28]. Thereby, the goal is to ensure that the parameters that are crucial for this goal are kept stable, whereas the less important parameters may vary on a larger scale. For the analysis of running in terms of the variability and stability of biomechanical parameters, different methods exist (e.g., uncontrolled manifold approach and tolerance noise covariation) and were applied in several studies [29,30,31]. Even though these methods are ideal for investigating the structure of motor variability in redundant solution spaces, they are not useful for distinguishing a target parameter [27,29,30]. Analyzing the variability characteristics of biomechanical parameters may help to understand which parameters are of high priority for the CNS [27,32]. Despite its relevancy, only a few studies have analyzed the variability in running kinematics [22,33,34,35,36]. 

To sum up, three key biomechanical parameters, i.e., vertical oscillation of CoM, stride frequency, and duty factor, have been shown to be closely related to running economy and running style. However, the influence of expertise level and running speed on the variability of these key biomechanical parameters has only been partially researched, even though they may reveal valuable information regarding the control of human locomotion. Therefore, the goal of this study was to investigate the effects of expertise level on vertical oscillation of CoM, stride frequency, and duty factor, as well as their variabilities at two different running speeds. It was hypothesized that expert runners would show lower mean values and lower variability of the considered parameters compared with novice runners, regardless of the running speed.

## 2. Materials and Methods

The data used in the present study were analyzed with the uncontrolled manifold approach as in a previous study [37]. 

### 2.1. Participants

The participants of this study were all male and comprised two groups: one group of 13 expert runners (EXP: age, 23.5 ± 3.6 years; height: 1.80 ± 0.06 m; weight, 66.8 ± 5.4 kg) and one group of 12 novice runners (NOV: age, 23.9 ± 3.8 years; height: 1.83 ± 0.07 m; weight, 72.2 ± 6.6 kg). EXP included runners with a 10 km personal best time below 35 min (32:59 ± 01:19 min) who had been members of a running club for at least 2 years (7.2 ± 3.2 years) and ran a minimum of 50 km per week (duration: 6.5 ± 1.7 h/week). NOV included runners who participated in a maximum of two sport sessions per week, including a maximum of one running session (0.2 ± 0.2 h/week). Importantly, this group only included runners who had never prepared for a running event or trained in a running club. It was important for all participants to be free of recent injuries or pain in the lower limbs. They were asked not to perform any intense workout on the day preceding the measurement. All participants provided their written informed consent prior to participation. This study was approved by the ethics committee of the Karlsruhe Institute of Technology.

### 2.2. Protocol

The study was performed on a motorized treadmill (h/p/cosmos Saturn; Nussdorf-Traunstein, Germany), with participants wearing a safety harness that was connected to an emergency off button. Before the measurements, a total of 22 anthropometric measurements were taken manually for each participant, and 41 markers were attached on their body according to the guidelines of the ALASKA (Advanced Lagrangian Solver in kinetic Analysis) modelling system [38]. After a standardized session of treadmill familiarization (6 min of walking and 6 min of running, [39,40], the speed of the treadmill was accelerated up to 15 km/h and held for 15 s in order for the participants to experience the running speed that they would run at during the measurements. After a 2 min break, the participants performed runs at speeds of 10 and 15 km/h in a counterbalanced order. The participants ran for approximately 1 min at each of the two speeds. For each speed, 3D marker data for the 41 markers were recorded during 20 consecutive gait cycles using 11 Vicon MX cameras (Vicon Motion Systems; Oxford Metrics Group, Oxford, UK) that were capable of recording at 200 Hz. The two running speeds for the measurements, 10 and 15 km/h, were chosen based on previous comparable studies [16,22]. In the pre-tests for this experiment, the chosen running speeds were confirmed to be proper.

### 2.3. Data Processing

The marker data were post-processed using the Vicon Nexus software (V1.8.5) and filtered with a 10 Hz low-pass Butterworth filter using Matlab (The MathWorks, Natick, MA, USA). The gait cycles were segmented using the foot marker data [41]. The CoM was estimated with the ALASKA Dynamicus modelling system [38]. To calculate osc_CoM, the difference between the minimum and the maximum height of the CoM (Equation (1)) was calculated for each of the 20 gait cycles. SF_norm and DF were calculated using previously published formulae [13]. To calculate SF_norm, SF was calculated as 60 divided by the sum of stance time and flight time, and then it was normalized to the leg length (Equation (2)). To calculate DF, the stance time was divided by twice the sum of the stance and flight time (Equation (3)). The variability of these parameters was calculated as the coefficient of variation (CV). Their mean values were included in the analysis to compare the results with those published in the existing literature.
osc_CoM = CoM_Zmax_ − CoM_Zmin_(1)
(2)$$\mathrm{SF}\_\mathrm{norm}=\frac{60}{\left(t_{stance}+t_{flight}\right)\cdot \sqrt{\frac{L_{0}}{g}}}\;$$
(3)$$\mathrm{DF}=\frac{t_{stance}}{2\cdot (t_{stance}+t_{flight})}\;$$

### 2.4. Statistical Analysis

Shapiro–Wilk tests were conducted to confirm the normality of data distribution. The dependent variables were osc_CoM, SF_norm, and DF, as well as their variabilities (operationalized by CV). For each of these dependent variables, a 2 × 2 repeated-measures ANOVA (rmANOVA) was calculated with the factors expertise level (EXP and NOV) and speed (10 and 15 km/h). In total, six rmANOVAs were performed. The Bonferroni–Holm method was used to correct the results for multiple comparisons [42]. The significance level was set a priori to *p* < 0.05. Partial eta-squared (small effect: $\eta_{p}^{2}$ < 0.06; medium effect 0.06 ≤ $\eta_{p}^{2}$ < 0.14; large effect: $\eta_{p}^{2}$ ≥ 0.14) was calculated as a measure of the effect size for rmANOVA.

## 3. Results

The results for the mean values and the variability are shown in Figure 1. 

### 3.1. Mean Value Changes in Vertical Oscillation, Normalized Frequency, and Duty Factor

The rmANOVA value for osc_CoM showed a non-significant main effect of the factor expertise level (*p* = 0.33, $\eta_{p}^{2}$ = 0.189) and a non-significant interaction effect of expertise level and speed, with a high effect size (*p* = 0.576, $\eta_{p}^{2}$ = 0.142). The main effect of the factor speed was also not significant (*p* > 0.999, $\eta_{p}^{2}$ = 0.054).

The rmANOVA for SF_norm showed a significant main effect of the factor speed, with a high effect size (*p* = 0.018, $\eta_{p}^{2}$ = 0.438) and a non-significant interaction between expertise level and speed (*p* > 0.999, $\eta_{p}^{2}$ = 0.001). The main effect of the factor expertise level was not significant (*p* > 0.999, $\eta_{p}^{2}$ = 0.014). In both groups, SF_norm increased from 10 to 15 km/h.

The rmANOVA for DF showed significant main effects of expertise level (*p* = 0.018, $\eta_{p}^{2}$ = 0.502) and speed (*p* = 0.018, $\eta_{p}^{2}$ = 0.908), with high effect sizes. However, the interaction effect of expertise level and speed was not significant (*p* > 0.999, $\eta_{p}^{2}$ = 0.008). In summary, DF decreased from 10 to 15 km/h and was overall higher for NOV than for EXP. 

### 3.2. Changes in Variability according to the Expertise Level at Two Running Speeds

With regard to the CV of osc_CoM, rmANOVA showed significant main effects of expertise level (*p* = 0.018, $\eta_{p}^{2}$= 0.792) and speed (*p* = 0.018, $\eta_{p}^{2}$ = 0.408), as well as non-significant interaction effects (*p* = 0.084, $\eta_{p}^{2}$ = 0.279), each with high effect sizes. This implies that the CV of osc_CoM was higher for NOV than for EXP. The CV of osc_CoM increased with an increase in speed.

The rmANOVA for the CV of SF_norm showed a significant effect of the factor expertise level, with a high effect size (*p* = 0.018, $\eta_{p}^{2}$ = 0.435). However, the effect of speed (*p* > 0.999, $\eta_{p}^{2}$ < 0.001) and the interaction effect of expertise level and speed (*p* > 0.999, $\eta_{p}^{2}$ = 0.025) were not significant. Accordingly, the results showed that the CV of SF_norm was higher for NOV than for EXP.

The rmANOVA for the CV of DF had no significant effects, but a high effect size was found for the factor expertise level (*p* = 0.520, $\eta_{p}^{2}$ = 0.155). In contrast, the effect sizes for speed (*p* > 0.999, $\eta_{p}^{2}$ = 0.017) and the interaction effect of expertise level and speed (*p* > 0.999, $\eta_{p}^{2}$ < 0.001) were low.

## 4. Discussion

The aim of this study was to analyze the effects of the expertise level on key biomechanical parameters and the variability of these parameters at two different running speeds. It was hypothesized that regardless of the running speed, expert runners are characterized by lower mean values and lower variability of all the considered parameters compared with their novice counterparts. The results indicated that the expert runners had a lower duty factor than the novices. Furthermore, the experts showed a significantly lower variability than the novices with regard to vertical oscillation of CoM and normalized stride frequency, independently of the running speed, but no differences in variability were observed for the duty factor. Based on the findings on this study, our hypotheses can be only partially accepted.

### 4.1. Lower Duty Factor for Expert Runners

EXP and NOV did not differ significantly in terms of osc_CoM (EXP, 10 km/h: 91.43 mm, 15 km/h: 95.06 mm; NOV, 10 km/h: 82.3 mm, 15 km/h: 81.36 mm). These findings are interesting, since more experienced runners are expected to have a better running economy [9,20], which was shown to be associated with a lower osc_CoM [2,8,9].

With regard to the results for SF_norm, they are in line with those of other studies [13,43], that is, an increased cadence was observed with an increase in speed. Based on these results, it can be suggested that SF_norm is not directly affected by the expertise level but, rather, is a function of the running speed.

DF, which describes the step time relative to the flight time, decreased with increasing speed in both groups in this study, and this finding is also in line with other published studies [8,44]. Furthermore, EXP showed an overall lower DF than NOV (10 km/h: 17.3% less; 15 km/h: 20.3% less); this indicates that EXP had a longer flight phase than NOV at a given stance time. The interpretation of the DF results is not straightforward, but an optimal level of DF seems to exist for runners. Even though a lower DF was shown to be associated with better running economy, a DF value that is too low could be uneconomical, given the high muscle activation that occurs over a very short stance time. Further, a DF value that is too high may indicate high start–stop accelerations and, therefore, a waste of mechanical work [45].

### 4.2. Lower Variability of Vertical Oscillation and Normalized Stride Frequency in Expert Runners

The results indicated that EXP had significantly lower variability in osc_CoM and SF_norm than NOV, independently of the running speed, whereas there were no significant changes in DF. In the time period over which a novice runner becomes an expert runner, a variety of changes occur in the runners’ body that range from physiological to neuromuscular adaptations that are necessary for movement efficiency [9,46]. Ultimately, the running economy is improved to decrease the total energy need, since humans inevitably tend to conserve energy from an evolutionary perspective [47] and prefer an energy-optimal gait [48]. The lower variability in osc_CoM and SF_norm found in EXP could mean that after multiple years of training, the CNS tries to reduce the variability in these parameters because they are important for the running economy as well as for a consistent running style [2,9,13]. However, it is important to note that it is difficult to directly draw this conclusion from the findings of this study. Even though the variability in these two parameters was lower in the EXP, it is possible that the CNS primarily controls other parameters which consequently influence the variability of these key parameters.

Differences in variability between the two running speeds, that is 10 and 15 km/h, were only detected for osc_CoM. These findings imply that the expertise level plays a major role in terms of variability in these parameters, whereas the effects of speed were rather small. This could, however, be dependent on the choice of the running speed. To the best of our knowledge, there are very few studies whose results can be compared with those of this study. One such study [22] reported that amateur runners showed a higher variability in stance time and SL at running speeds of 15–16 km/h than at a speed of 10 km/h. However, it analyzed different parameters from those analyzed in this study and calculated the standard deviation (SD) instead of the CV. Therefore, it is difficult to compare their findings with the present findings. In this study, CV was preferred over SD, since it reflects changes in SD normalized to the mean value. It is also important to note that regardless of the research question, it is difficult to analyze the effects of the running speed in terms of running economy and running style, since humans seem to have an energy-optimal gait and prefer to move at this optimal speed, thus minimizing the energy requirement [48].

### 4.3. Limitations

Our study has some limitations that should be considered when interpreting the results. The first limitation is that the focus was on the running economy, even though it was not quantified by the parameters that are usually used for the operationalization of the running economy (e.g., VO_2_ [12]). Rather, the key biomechanical parameters that have been shown to be strongly related to running economy and running style [2,8,9,13] were used. Another limitation of this study is that the measurements were conducted with a treadmill under laboratory conditions, which differ from the usual environment that most runners are exposed to. On the other hand, the use of a treadmill enabled a precise control of speed and, therefore, eliminated any confounding effects caused by a variable running speed. In addition, similar studies have also been performed under laboratory conditions [35,49]. The third limitation is that the sample size was chosen on the basis of comparable studies [20,50,51]. It might have been better to choose the sample size based on an a priori power analysis. The fourth and final limitation is that the chosen speeds were the same for all participants. It would have been preferable to choose individual speeds based on individual thresholds, since a connection between running speed and running economy in terms of VO_2_ max has been demonstrated [9]. However, the interpretation of the results would have been much more complicated due to the addition of speed as an individually varying factor. Despite this, the results do indicate an overall lower variability among EXP than NOV, independent of the speed that they were running at. 

## 5. Conclusions

The aim of this study was to investigate the effects of the expertise level on key biomechanical parameters and their variabilities at two different running speeds. The findings showed that, independently of the running speed, expert runners had a lower duty factor and showed less variable vertical oscillation of CoM and normalized stride frequency than novice runners, but the variation in duty factor did not differ between the two groups. At a higher running speed, the variability in vertical oscillation of CoM was higher, whereas the variability in the other two parameters did not change significantly, independently of the expertise level. Based on these results, it can be suggested that two of the three considered parameters, i.e., vertical oscillation of CoM and normalized stride frequency, are important parameters, whose variability decreased with the increase in the expertise level. A lower variability of these parameters found in expert runners may indicate that after multiple years of training, the CNS tries to reduce the variability in these parameters because they are important for the running economy as well as for a consistent running style. Further studies should address the variability of key biomechanical parameters in terms of running economy and running style in a more detailed manner to identify the parameters that are of high priority for the CNS.

## Figures and Tables

**Figure 1 bioengineering-09-00616-f001:**
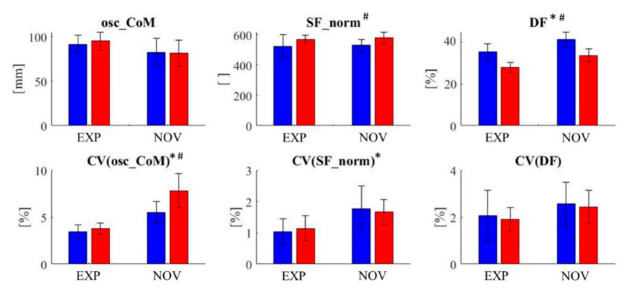
Mean value (**top row**) and coefficient of variation (CV) (**bottom row**) for the three parameters vertical oscillation of CoM (osc_CoM), normalized stride frequency (SF_norm), and duty factor (DF). The error bars show the standard deviation of the respective values. The values for the 10 km/h condition are shown in blue, and the values for the 15 km/h condition are shown in red; * significant effect for the factor expertise level, # significant effect for the factor speed.

## Data Availability

The data presented in this study are available on request from the corresponding author.

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
