# Peer review of "Changes in Key Biomechanical Parameters According to the Expertise Level in Runners at Different Running Speeds"

_bioengineering, 2022, doi:10.3390/bioengineering9110616_

Round 1
Reviewer 1 Report
Firstly, I would like to thank the Editors from Bioengineering for giving me the opportunity to have reviewed the manuscript ID: bioengineering-1963655 titled “Changes in key biomechanical parameters according to expertise level in runners at different running speeds”. The main purpose of the present study under consideration was to investigate the effects of expertise level on osc_CoM, SF_norm, and DF, as well as their variabilities at two different running speeds. Working hypothesis was that expert runners would have a lower osc_CoM, SF_norm, and DF than novice runners and show a higher variability in all the considered parameters, regardless of running speed. 13 expert runners (EXP) and 12 novice runners (NOV) participated. The work covers an interesting research topic, main findings revealed that EXP presented a higher osc_COM and a lower DF than NOV; EXP had also less variability in osc_CoM and SF_norm than NOV, regardless of running speed, while DF variability was similar between groups. As such, I have major/minor specific suggestions that need to be addressed before I can make a final decision on the acceptability of the present manuscript to publication:
P3L109. “To sum up, osc_COM SF_norm and DF have been shown to be closely related to running economy and running style. However, the influence of expertise level and running speed on the variability of these key biomechanical parameters has only been partially researched, even though they may reveal valuable information regarding the control of human locomotion. Therefore, the goal of this study was to investigate the effects of expertise level on osc_CoM, SF_norm, and DF, as well as their variabilities at two different running speeds. It was hypothesized that expert runners would have a lower osc_CoM, SF_norm, and DF than novice runners and show a higher variability in all the considered parameters, regardless of running speed.” At this point I would like to urge the authors to avoid acronyms in the most important study moments (e.g. aim, first paragraph of discussion and conclusions). It will likely help improve the readability of the present manuscript.
P3L124. “The participants of this study were all male and comprised two groups: one group of 13 expert runners (EXP: age, 23.5 ± 3.6 years; height: 1.80 ± 0.06 m; weight, 66.8 ± 5.4 kg) and one group of 12 novice runners (NOV: age, 23.9 ± 3.8 years; height: 1.83 ± 0.07 m; 126 weight, 72.2 ± 6.6 kg)”. It is necessary to insert sample size calculation, in order to justify the number of runners recruited, in each of the experimental groups, to reach sufficient statistical power in the comparisons performed.
P4L152. “These two speeds were chosen because 10 km/h and 15 km/h are considered to be comfortable speeds for NOV and EXP, respectively.”. Please expand rationale for adopting such velocities in the analysis performed. Currently it is too vague.
P4L174. “In total, six rmANOVAs were performed.”. Consider whether it is necessary to adjust p-values for multiples comparisons performed.
P5L188. “Trends in vertical oscillation, normalized frequency, and duty factor.” Please clarify “trends” or consider delete it. Unfortunately, trends are of poor value, otherwise insert effect sizes and then attempt to discuss non-significant moderate-to-large differences.
P5L223. “In previous studies, osc_COM, SF_norm, and DF were suggested to be closely related to running economy and running style [2,9,13]. Even though analyzing the variability characteristics of the parameters may reveal valuable information about the control of human locomotion, the variability of these key biomechanical parameters has not yet been investigated. Accordingly, the aim of this study was to analyze the effects of expertise level on the key biomechanical parameters and their variabilities at two different running speeds. It was hypothesized that regardless of running speed, EXP have lower osc_CoM, SF_norm, and DF and show lower variability for all the considered parameters than NOV”. Please rephrase the first paragraph of discussion using the following structure: 1) repeated study aim; 2) main findings and 3) brief direction of discussions.
P7L312. Please revise the whole conclusion section: avoid acronyms and provide a clear answer to the study question, avowing an exhaustive repetition of study results here.
Author Response
The authors thank the reviewer for the critical revision of our work and for pointing out the important aspects which were not clear in the manuscript. We wrote our answers below each comment separately and marked the changes with yellow in our manuscript. We hope that we addressed all of the comments adequately.
Firstly, I would like to thank the Editors from Bioengineering for giving me the opportunity to have reviewed the manuscript ID: bioengineering-1963655 titled “Changes in key biomechanical parameters according to expertise level in runners at different running speeds”. The main purpose of the present study under consideration was to investigate the effects of expertise level on osc_CoM, SF_norm, and DF, as well as their variabilities at two different running speeds. Working hypothesis was that expert runners would have a lower osc_CoM, SF_norm, and DF than novice runners and show a higher variability in all the considered parameters, regardless of running speed. 13 expert runners (EXP) and 12 novice runners (NOV) participated. The work covers an interesting research topic, main findings revealed that EXP presented a higher osc_COM and a lower DF than NOV; EXP had also less variability in osc_CoM and SF_norm than NOV, regardless of running speed, while DF variability was similar between groups. As such, I have major/minor specific suggestions that need to be addressed before I can make a final decision on the acceptability of the present manuscript to publication:
P3L109. “To sum up, osc_COM SF_norm and DF have been shown to be closely related to running economy and running style. However, the influence of expertise level and running speed on the variability of these key biomechanical parameters has only been partially researched, even though they may reveal valuable information regarding the control of human locomotion. Therefore, the goal of this study was to investigate the effects of expertise level on osc_CoM, SF_norm, and DF, as well as their variabilities at two different running speeds. It was hypothesized that expert runners would have a lower osc_CoM, SF_norm, and DF than novice runners and show a higher variability in all the considered parameters, regardless of running speed.” At this point I would like to urge the authors to avoid acronyms in the most important study moments (e.g. aim, first paragraph of discussion and conclusions). It will likely help improve the readability of the present manuscript.
Thank you very much for this critical comment. We revised our whole manuscript accordingly to keep the use of acronyms at minimum and maximize the readability in the most important parts. We hope now one can follow our manuscript better.
P3L124. “The participants of this study were all male and comprised two groups: one group of 13 expert runners (EXP: age, 23.5 ± 3.6 years; height: 1.80 ± 0.06 m; weight, 66.8 ± 5.4 kg) and one group of 12 novice runners (NOV: age, 23.9 ± 3.8 years; height: 1.83 ± 0.07 m; 126 weight, 72.2 ± 6.6 kg)”. It is necessary to insert sample size calculation, in order to justify the number of runners recruited, in each of the experimental groups, to reach sufficient statistical power in the comparisons performed.
Thank you for your comment. The number of participants was determined based on common sample size and it is even higher than in comparable studies (e.g., n=10-12 (Garciá-Pinillos et al., 2019; n=10 (de Ruiter et al., 2014)). Further, the effect sizes were high (operationalized by eta squared) in case of the significant changes. Nevertheless, it would have been better, if we had performed a priori power-analysis to determine the minimum required number of participants. Therefore, we added this point under our limitations (Line 310 ff.).
Garciá-Pinillos, F., Garciá-Ramos, A., Ramírez-Campillo, R., Latorre-Román, P. & Roche-Seruendo, L. E. How Do Spatiotemporal Parameters and Lower-Body Stiffness Change with Increased Running Velocity? A Comparison between Novice and Elite Level Runners. J. Hum. Kinet. 70, 25–38 (2019).
de Ruiter, C. J., Verdijk, P. W. L., Werker, W., Zuidema, M. J. & de Haan, A. Stride frequency in relation to oxygen consumption in experienced and novice runners. Eur. J. Sport Sci. 14, 251–258 (2014).
P4L152. “These two speeds were chosen because 10 km/h and 15 km/h are considered to be comfortable speeds for NOV and EXP, respectively.”. Please expand rationale for adopting such velocities in the analysis performed. Currently it is too vague.
Thank you for pointing this out. The running velocities 10 km/h and 15 km/h were typically used in various comparable studies (e.g., Gómez-Molina et al., 2017; García-Pinillos et al., 2020). In the pre-tests for this experiment, the chosen running speeds were confirmed to be proper. We revised this part in our method section to clarify it.
Gómez-Molina J, Ogueta-Alday A, Stickley C, Cámara J, Cabrejas-Ugartondo J, García-López J. Differences in Spatiotemporal Parameters Between Trained Runners and Untrained Participants. J Strength Cond Res. 2017;31(8):2169-2175. doi:10.1519/JSC.0000000000001679
García-Pinillos F, Jerez-Mayorga D, Latorre-Román P, Ramirez-Campillo R, Sanz-López F, Roche-Seruendo LE. How do Amateur Endurance Runners Alter Spatiotemporal Parameters and Step Variability as Running Velocity Increases? A Sex Comparison. J Hum Kinet. 2020;72(1):39-49. doi:10.2478/hukin-2019-0098
P4L174. “In total, six rmANOVAs were performed.”. Consider whether it is necessary to adjust p-values for multiples comparisons performed.
Thank you very much for this critical comment. We corrected our p-values following the Bonferroni Holm method. After correction, the previously significant effects remained significant mostly. For the two exceptional cases, we made necessary changes in our manuscript.
P5L188. “Trends in vertical oscillation, normalized frequency, and duty factor.” Please clarify “trends” or consider delete it. Unfortunately, trends are of poor value, otherwise insert effect sizes and then attempt to discuss non-significant moderate-to-large differences.
Thank you for this comment. The word “trend” was misleading. We had used this to point out that the mean values differed significantly and with high effect sizes. We changed it as “mean value changes” (Line 197)
P5L223. “In previous studies, osc_COM, SF_norm, and DF were suggested to be closely related to running economy and running style [2,9,13]. Even though analyzing the variability characteristics of the parameters may reveal valuable information about the control of human locomotion, the variability of these key biomechanical parameters has not yet been investigated. Accordingly, the aim of this study was to analyze the effects of expertise level on the key biomechanical parameters and their variabilities at two different running speeds. It was hypothesized that regardless of running speed, EXP have lower osc_CoM, SF_norm, and DF and show lower variability for all the considered parameters than NOV”. Please rephrase the first paragraph of discussion using the following structure: 1) repeated study aim; 2) main findings and 3) brief direction of discussions.
Thank you for this critical comment. We revised the first paragraph of our discussion section accordingly. We hope now it is clearer and better structured.
P7L312. Please revise the whole conclusion section: avoid acronyms and provide a clear answer to the study question, avowing an exhaustive repetition of study results here.
Thank you very much for pointing this out. We revised our conclusion section to avoid the acronyms and make the answer to our research question clearer. We also deleted the unnecessary repetitions.
Reviewer 2 Report
The paper presents important news about the level of expertise of street runners on economy parameters in running at two chosen speeds.
Data were collected according to well-defined criteria, and analyzed from the perspective of the literature on the subject, in addition to determining in the text the limitations of the article under the data collected and how the authors projected the results.
The discussion is very good, having been done in topics to facilitate the observation of readers, showing what is new in the paper presented.
My suggestion is to accept the paper as is it.
Author Response
The authors thank the reviewer for the critical revision of our work and for the kind feedbacks.
Reviewer 3 Report
This paper about economics of running at two speeds with experienced and nover runners is well written and gives innterestng results. However the two groupsare small and therefore the many non significant differences can be caused.
The abstract is not clear about the two speeds that were used and also the results are not clear explained and als not the Duty factor.
What is CV in figure 1?
Author Response
The authors thank the reviewer for the critical revision of our work and for pointing out the important aspects which were not clear in the manuscript. We wrote our answers below each comment separately and marked the changes with yellow in our manuscript. We hope that we addressed all of the comments adequately.
This paper about economics of running at two speeds with experienced and nover runners is well written and gives innterestng results. However the two groupsare small and therefore the many non significant differences can be caused.
The number of participants was determined based on comparable studies and it is even higher than in comparable studies (e.g., n=10-12 (Garciá-Pinillos et al., 2019; n=10 (de Ruiter et al., 2014)). Further, the effect sizes were high (operationalized by eta squared) in case of the significant changes and mostly small for the non-significant changes. Nevertheless, it would have been better, if we had performed a priori power-analysis to determine the minimum required number of participants. Therefore, we added this point under our limitations (Line 310 ff.).
Garciá-Pinillos, F., Garciá-Ramos, A., Ramírez-Campillo, R., Latorre-Román, P. & Roche-Seruendo, L. E. How Do Spatiotemporal Parameters and Lower-Body Stiffness Change with Increased Running Velocity? A Comparison between Novice and Elite Level Runners. J. Hum. Kinet. 70, 25–38 (2019).
de Ruiter, C. J., Verdijk, P. W. L., Werker, W., Zuidema, M. J. & de Haan, A. Stride frequency in relation to oxygen consumption in experienced and novice runners. Eur. J. Sport Sci. 14, 251–258 (2014).
The abstract is not clear about the two speeds that were used and also the results are not clear explained and als not the Duty factor.
Thank you for your comment. We revised our abstract to make it clearer. We hope it became better.
What is CV in figure 1?
Thank you very much for pointing this out. CV is coefficient of variation. We added it into the caption of the figure as well.
Round 2
Reviewer 1 Report
I would like to thank the authors for addressing all my comments. The paper reads much better and can be now recommended for publication in its current form.